# Interference on Iodine Uptake and Human Thyroid Function by Perchlorate-Contaminated Water and Food

**DOI:** 10.3390/nu12061669

**Published:** 2020-06-04

**Authors:** Giuseppe Lisco, Anna De Tullio, Vito Angelo Giagulli, Giovanni De Pergola, Vincenzo Triggiani

**Affiliations:** 1ASL Brindisi, Unit of Endocrinology, Metabolism & Clinical Nutrition, Hospital “A. Perrino”, Strada per Mesagne 7, 72100 Brindisi, Puglia, Italy; g.lisco84@gmail.com; 2Interdisciplinary Department of Medicine—Section of Internal Medicine, Geriatrics, Endocrinology and Rare Diseases, University of Bari “Aldo Moro”, School of Medicine, Policlinico, Piazza Giulio Cesare 11, 70124 Bari, Puglia, Italy; annadetullio16@gmail.com (A.D.T.); vitogiagulli58@gmail.com (V.A.G.); 3Clinic of Endocrinology and Metabolic Disease, Conversano Hospital, Via Edmondo de Amicis 36, 70014 Conversano, Bari, Puglia, Italy; 4Department of Biomedical Sciences and Human Oncology, Section of Internal Medicine and Clinical Oncology, University of Bari Aldo Moro, Piazza Giulio Cesare 11, 70124 Bari, Puglia, Italy; gdepergola@libero.it

**Keywords:** perchlorate, Natrium/Iodide symporter, iodine, endocrine disruptors, review, drinking and Food, Hypothyroidism

## Abstract

Background: Perchlorate-induced natrium-iodide symporter (NIS) interference is a well-recognized thyroid disrupting mechanism. It is unclear, however, whether a chronic low-dose exposure to perchlorate delivered by food and drinks may cause thyroid dysfunction in the long term. Thus, the aim of this review was to overview and summarize literature results in order to clarify this issue. Methods: Authors searched PubMed/MEDLINE, Scopus, Web of Science, institutional websites and Google until April 2020 for relevant information about the fundamental mechanism of the thyroid NIS interference induced by orally consumed perchlorate compounds and its clinical consequences. Results: Food and drinking water should be considered relevant sources of perchlorate. Despite some controversies, cross-sectional studies demonstrated that perchlorate exposure affects thyroid hormone synthesis in infants, adolescents and adults, particularly in the case of underlying thyroid diseases and iodine insufficiency. An exaggerated exposure to perchlorate during pregnancy leads to a worse neurocognitive and behavioral development outcome in infants, regardless of maternal thyroid hormone levels. Discussion and conclusion: The effects of a chronic low-dose perchlorate exposure on thyroid homeostasis remain still unclear, leading to concerns especially for highly sensitive patients. Specific studies are needed to clarify this issue, aiming to better define strategies of detection and prevention.

## 1. Introduction

Endocrine disrupting chemicals (EDCs) have been defined as a group of compounds or a mixture of natural or man-housed exogenous chemicals which interfere with the hormonal network, or induce endocrine cell damage [1]. Interference may be attributable to several mechanisms such as receptor agonism or antagonism, modulation of receptor expression, modification of signal transduction, hormone synthesis or incretion, plasmatic distribution and clearance [2]. Moreover, epigenetic effects have been hypothesized for EDCs and concerns about a possible “transmission” of EDCs across the generations is a topic of debate [3,4]. To date, a wide range of environmental chemicals have been identified as being involved in the pathogenesis of thyroid diseases [5,6] and several chemicals or common pollutants may act as thyroid disruptors [7,8,9]. Perfluorooctanoic acid [10], a chemical largely employed for the manufacturing of waxes, cosmetics, carpets, cleaning or waterproof products, and bisphenols [11], hugely used as plasticizers, were found to increase the prevalence of thyroid diseases in exposed patients [12], including thyroid autoimmunity [13]. Moreover, legacy pesticides were experimentally shown to affect thyroid function [14] and, despite some controversy, they may also induce hypothyroidism, thyroid autoimmunity, thyroid volume enlargement or nodules in humans [15]. The bactericide triclosan was mostly found in personal hygiene products (oral care, shampoos, hand sanitizers, soaps), and was proven to increase the risk of thyroid diseases, too [16]. Thyroid disruption includes different pathways, and may be due to either interference or synergism among different EDCs [17]. The leading mechanisms of thyroid interference by pollutant agents have been explored, and frequently include the inhibition of thyroperoxidase activity, competitive natrium-iodide symporter (NIS) inhibition, impairment of binding protein transport and peripheral deiodinase activity, enhancement of liver catabolism [18]. Since food and drinks are also a relevant source of thyroid disruptors, a lifelong human exposure to these chemicals could induce potentially harmful consequences on thyroidal homeostasis. Given this consideration, this review aims to specifically focalize on NIS interference by specific agents, mainly perchlorate compounds, which are commonly found in food and drinks.

## 2. Materials and Methods

The authors summarized iodine metabolism and its importance in thyroid homeostasis and hormonal synthesis. Furthermore, the authors searched PubMed/MEDLINE, Scopus, Web of Science, institutional websites and Google for relevant information about the fundamental mechanism of NIS interference induced by perchlorate compounds orally assumed and the consequences on thyroidal health status associated with chronic exposure to these chemicals.

## 3. Results

### 3.1. Overview on Iodine Metabolism in Healthy Humans

The primary source of iodine (I) is represented by natural food (seafood, milk, eggs, vegetables, legumes, fruits), fortified food (salt) and mineral waters. I is basically available in two forms, organic and inorganic (iodide); the latter form is absorbed at the level of stomach and duodenum [19] through a specific natrium-iodide symporter (NIS) which regulates iodine homeostasis in human body [20]. After gastrointestinal absorption, I enters the circulation, undergoing to a large distribution into the plasma, red blood cell cytoplasm and extracellular fluid, and is finally intercepted by tissues [21]. A wide range of tissues express the NIS, including salivary glands, breast, and thyroid [22]. Nevertheless, thyroid represents the most important reservoir of the ion considering that, in a healthy human body, the gland normally stores up to 80% of the entire iodine pool (15–20 mg). The NIS is a 13-domain transmembrane protein which mediates transmembrane I and sodium (1 to 2 ratio) transport at the level of thyrocyte’s basolateral membrane [23,24]. Transmembrane sodium gradient is generated by the sodium-potassium ATPase pump which indirectly provides energy for an almost continuous intrathyroidal I uptake (secondary active transport). Given this thyroid avidity, I concentration in thyrocytes is 30 to 60 times higher than its plasmatic levels [25]. As a mean, thyroid secretes 80 µg a day of I in the form of both levothyroxine and triiodothyronine [26]. Due to peripheral metabolism of thyroid hormones, I circulates in bloodstream finally undergoing to both renal and hepato-biliary clearance and thyroidal re-uptake, as well. An intrathyroidal I recycling has also been described [27]. Thyroidal uptake considerably fluctuates according to I intake, and ranges from 10% to over 80% of the entire amount of ingested I. Contrariwise, urinary I excretion is inversely correlated with thyroid uptake, and in the case of adequate I intake, more than 90% of the ion is cleared by the kidneys with urine [28]. Urinary I concentration is thus a reliable biomarker of I intake, and is a useful tool for screening patients suspected for I deficiency [25]. Both the thyroid hormone synthesis and urinary I excretion increase during pregnancy [29], while 126 to 269 µg of I could be excreted with each liter of breast milk in lactating women [30]. Iodine intake is generally recommended at 150 µg per day for adults in order to ensure the daily iodine recycle [31]. Thus, the recommended dose of iodine intake raises at 200 - 250 µg per day during pregnancy and lactation for sustaining an increased requirement [32]. I is an essential micronutrient for thyroid hormones synthesis [33,34]. Afterward the transition into thyrocyte cytoplasm, I moves towards the apical surface of thyrocyte’s plasmatic membrane into the follicular lumen. This transport is mediated by a ionic carrier belonging to the SLC26A family, otherwise known as pendrin [35], and is also expressed at the level of the inner ear, kidney and bowel. Specifically, pendrin is essential for favoring the efflux of iodine into follicular space in exchange of chloride (1 to 1 ratio) and a defective synthesis or function of this carrier is responsible for a the so called Pendred’s syndrome [36]. Once into follicular lumen, I undergoes oxidation by thyroperoxidase, thus becoming promptly available for thyroglobulin’s organification. Thyroid I content is the most important regulator of thyroid hormone synthesis. Indeed, I overload reduces the expression of NIS, decreases both the thyroid peroxidase and deiodinases activities, and finally leads to a transient impairment of thyroid hormone synthesis [37]. In predisposed patients, iodine excess increases oxidative stress, and may induce or exacerbate thyroid autoimmunity and hypothyroidism (Wolff–Chaikoff effect) [38]. Finally, I overload may exacerbate a latent hyperthyroidism in patient with single thyroid nodule or multinodular goiter [39].

### 3.2. Perchlorate Compounds and Iodine Interference 

The evidence that high doses of perchlorate (ClO_4_^−^) anion decreased thyroid hormone synthesis has been known since the 1950s [40], and given this peculiarity, it has been used to effectively treat hyperthyroidism such as in Graves’ disease and amiodarone-induced hyperthyroidism [41]. Specifically, ClO_4_^−^ competes with I at the level of the NIS (Figure A1), the former having a 30-fold higher affinity for the symporter when compared to the latter [42]. A dose-response sigmoid curve has been reported for describing NIS sensitivity to ClO_4_^−^ inhibition in different species and the half maximal inhibiting concentration in humans was found at 1.566 μM [43]. To confirm these experimental results, an orally delivered acute exposure to up to 520 µg/kg of body weight (bw) induced a significant increase in serum thyroid stimulating hormone (TSH) levels, with a relevant decline in serum-free levothyroxine concentrations [44]. On the other hand, it is thought that a chronic low-dose exposure to ClO_4_^−^, normally observed as the consequence of food and drink intake, could impair thyroid function by reducing iodine uptake particularly in predisposed individuals, such as those with an underlying iodine deficiency [45,46].

### 3.3. Perchlorate Compounds in Food and Water

ClO_4_^−^ may naturally occur in the atmosphere from spontaneous photogenic reaction between chloride and ozone, or arises from man-made products such as oxidizers, fertilizers, explosives, propellants, fireworks, airbag inflators spread into environment. In addition, ClO_4_^−^ can be also produced from the degradation of the common water disinfectant hypochlorite [47]. Perchlorate compounds occur in different form, such as metal perchlorate, ammonium and alkali metal forms, organic and inorganic forms and salts. Antarctic ice represents the most important sediment of ClO_4_^−^ in the planet, with different concentrations depending on drilling areas [48]. The Atacama desert (Chile, South America) is another important natural source of geogenic ClO_4_^−^, and elevated concentrations of its compounds have been found in soil (290 to 2565 µg/Kg) and surface waters (744 to 1480 µg/L) [49]. Other relevant sources of natural ClO_4_^−^ have been discovered in Alaska, Puerto Rico, New Mexico, Texas, California (United States of America, USA), and Bolivia (South America). Anthropogenic ClO_4_^−^ compounds have been found in soil, sea and rainwater, surface and groundwaters, indoor and outdoor dust, ice and snow [50]. Given data from ice drilling analyses, anthropogenic ClO_4_^−^ started to accumulate in Arctic ice from the 1980s [51]. In Devon Island (Canada, North America), ClO_4_^−^ compounds were found in ice and snow at variable concentrations ranging from 1 to 18 ng/L [52]. A great variability in rainwater ClO_4_^−^ levels was observed due to differences in analyzed geographical sites and seasonality. In fact, ClO_4_^−^ concentration ranges from 0.02 to 1.6 µg/L in Texas (USA) [53], 0.02 to 6.9 µg/L in India (Asia) [54], and 0.35 to 27.3 ng/mL in China (Asia) [55]. Moreover, Munster et al. evaluated the levels ClO_4_^−^ in total deposition from November 2005 to July 2007 in Long Island (New York State), relieving a mean concertation of 0.21 µg/L and with a maximum level of 2.81 µg/L recorded after fireworks displays occurred during the Independence day celebration [56]. Another observation reported different levels of ClO_4_^−^ in wet deposition only, ranging from <5 to 105 ng/L (mean 14 ng/L) with higher concentrations recorded in spring and summer than winter [57]. Soil usually does not retain ClO_4_^−^ and more than 90% is confined in the aqueous phase [58] where ClO_4_^−^ spreads and persists due to its high solubility and resistance to photolysis and anaerobic bacterial biodegradation [59]. Fruits and vegetables represent a relevant food source of ClO_4_^−^, particularly because of the widespread use of perchlorate-based fertilizers [60]. In particular, leafy vegetables, spinach, salad plants, raspberries, apricots, asparagus, cantaloupes, and tomatoes accumulate ClO_4_^−^ as a consequence of farming techniques [61]. The mean concentration of perchlorate in tested food appears variable and the highest levels have been found in Guatemalan cantaloupes (156 µg/Kg), spinach (133 µg/Kg), Chilean green grapes (45.5 µg/Kg) and Romaine lettuce (29 µg/Kg) [62]. Vega et al. reported variable concentration of ClO_4_^−^ in Chilean drinking waters which ranged from 4 to 120 µg/L [63]. Conversely, lower levels of ClO_4_^−^ in drinking water have been observed in the USA [64] and Europe, including Italy (0.5–75 µg/L) [65].

### 3.4. Chronic Esposure to Perchlorate Compounds by Food and Drinking Water

The 2018 “Italian Institute for Food and Agriculture Market Services” ranking reported the USA as the most valuable country in exporting fruits and vegetables, followed by Mexico and Chile. Given the volume of exports, Spain (4th) and Italy (7th) are responsible for the 42% of the entire European market of fruits and vegetables, ahead of Poland, France and Greece [66]. Chile has a remarkable export economy [67], and usually exports several thousands of millions of kilograms of fruits a year worldwide [68]. Specifically, the European Union is Chile’s third-largest trade partner in the world, after China and the USA, and currently imports 19% of the Chilean global export of vegetable products [69]. Cherries and table grapes, followed by apples, Chilean blueberries and plums are the most exported vegetable products to Europe. Vegetables from Chile are notoriously rich in ClO_4_^−^ and the excessive consumption of these products could have chronically negative consequences on thyroid homeostasis. Indeed, ClO_4_^−^ food exposure is essentially driven by vegetables and fruits and widely ranged according to geographical area as well as seasonality [70]. To confirm this assumption, ClO_4_^−^ was detected in a wide range of vegetable samples, ranging from 21 to 162 μg/kg [71]. Vegetables consumption in Italy seems to slightly but continuously increase over time and some of the most consumed vegetable products, such as spinach, leaf vegetables and spices were found to be a relevant source of ClO_4_^−^ [72]. Normal consumption of these vegetables does not usually lead to exceeding the maximal total daily dose according to the European Food Safety Authority 2014 (0.3 µg/Kg of bw). However, a higher daily consumption of these products led to a relevant exceeding of the maximal tolerated dose by 32% in adults, 61% in children and 56% in infant [72]. In addition, tea and herbal infusions could represent another relevant source of ClO_4_^−^, oscillating from 630 to 730 µg/Kg for dark tea; 80 to 430 µg/Kg for black tea; and 250 to 500 µg/Kg for green tea [73]. Therefore, the consumption of the aforementioned products should be moderate and intermittent for avoiding a consistent ClO_4_^−^ overload. Indeed, acute exposure to high or very high levels of ClO_4_^−^ normally is not enough for overcoming thyroidal compensation and ability to maintain normal serum concentration of thyroid hormones in healthy individuals [64]. Chronic consumption of ClO_4_^−^ in adults has been estimated as high as 0.07 to 0.34 µg/Kg of body weight per day in Europe [70], and 0.2 to 0.4 µg/Kg of body weight per day in the USA [74]. Despite ClO_4_^−^ consumption being generally below the level of recommended reference dose in adults [75], it may become critical, especially in some categories, such as children, high sensitive patients, cigarette smokers, iodine deficient people, and pregnant and breast feeding women as well [76,77,78]. Indeed, the inhibition of I uptake and any potential downstream effects induced by ClO_4_^−^ are strictly dependent on the exposure to other environmental NIS inhibitors, such as thiocyanates and nitrates, and iodine intake itself [79]. These potential confounders should therefore be considered in future studies and calculations for risk assessment [80]. Finally, breast milk and infant formulas are the most significant sources of ClO_4_^−^ for newborns and infants [81,82,83]. Compared with adults, infants and children exhibited a greater ClO_4_^−^ exposure per Kg of bw per day [75,84], particularly breastfed children (0.22 µg/Kg of bw/day) respective to those fed by cow milk-based formula (0.1 µg/Kg of bw/day) or soy-based formula (0.027 µg/Kg of bw/day) [85]. Food intake more than drinking water is considered the main source of ClO_4_^−^ for children [81] and adults [86], since ClO_4_^−^ exposure from drinking water alone is not able to suppress thyroid function [87]. Nevertheless, this assumption is controversial considering that other results suggest opposite conclusions [70,74].

### 3.5. Perchlorate Compounds Toxicity

From this point of view, concerns have been supposed in case of ClO_4_^−^ exposure during fetal and infantile life [88,89]. The placental NIS ensures maternal-to-fetal transition of I [90], therefore allowing fetal uptake of ClO_4_^−^ and other goitrogen chemicals, too. Blount et al. specifically analyzed the perinatal exposure to goitrogen chemicals in 150 mothers from New Jersey (USA), showing that the placental barrier was more permeable to I respective to goitrogens and maternal urinary ClO_4_^−^ concentrations were directly correlated with ClO_4_^−^ concentration in amniotic fluid, thus resulting an useful tool for assessing fetal exposure [91]. As observed in a Chinese population, ClO_4_^−^ was detected in infant’s urine (22.4 ng/mL) and cord blood serum (3.2 ng/mL) at a concentration about 22 times greater compared to that reported by Blount (0.14 µg/L) [92]. This finding is difficult to explain, but could be attributable to different environmental exposures or dissimilarities in assay or both. Several studies analyzed the impact of a mild-to-moderate exposure to ClO_4_^−^ in early pregnancy on both maternal thyroid function and several neonatal outcomes. In a cross-sectional trial in Athens (Greece), 139 first-trimester pregnant women with mild iodine deficiency were chronically exposed to dietary sources of ClO_4_^−^ as suggested by median levels of urinary ClO_4_^−^ concentration at around 4 µg/L. The authors specifically found that ClO_4_^−^ urinary concentration, possibly associated with a moderate iodine deficiency, was inversely related with plasmatic levels of triiodothyronine and thyroxin in this cluster of patients [93]. A cross-sectional study in 200 first-trimester Thai pregnant women (<14 weeks of gestation age) confirmed a chronic low-level environmental exposure to ClO_4_^−^ compounds (and thiocyanates) and this exposure was positively associated with serum TSH concentration and negatively related with serum levothyroxine levels [94]. Data from San Diego (South California) reported a mean urinary ClO_4_^−^ concentration of 8.5 µg/L in first-trimester pregnant women, and the higher the level of ClO_4_^−^, the higher the level of TSH and the lower those of total thyroxine and free thyroxine [95]. Pearce et al. analyzed the effects of environmental exposure to ClO_4_^−^ in a cohort of 1600 first-trimester pregnant women, with mild-to-moderate iodine deficiency, who had been enrolled in the Controlled Antenatal Thyroid Screening Study (CATS) from Cardiff (Wales) and Turin (Italy). The results of this observation displayed a low-level environment exposure to ClO_4_^−^ in all participants but no thyroidal impairment due to this contamination was noted [96]. These findings were also confirmed in first-trimester pregnant women from Los Angeles (California) and Cordoba (Argentina) in whom a low concentration of urinary ClO_4_^−^ were detected (mean of 7.8 and 13.5 µg/L, respectively) but no correlation with ClO_4_^−^ exposure and thyroid function was demonstrated [97]. A cross-sectional association between urinary ClO_4_^−^, thiocyanate and nitrate concentration and thyroid function was also assessed in healthy pregnant women living in New York City (New York State). The results confirm that a co-occurrent exposure to ClO_4_^−^, thiocyanate and nitrate may possibly impair thyroid homeostasis leading to hypothyroidism and ClO_4_^−^ specifically displayed the largest weight in driving this outcome [98]. Taylor et al. evaluated the relationship between maternal ClO_4_^−^ exposure and neurocognitive development in first-trimester pregnant women with hypothyroidism or hypothyroxinemia and mild iodine deficiency. The results display that maternal urinary ClO_4_^−^ concentration in the highest 10% of the population were associated with an higher risk of offspring’s verbal intellective quotient impairment [odds ratio 3.14 (1.38–7.13), p 0.006] and levothyroxine replacement did not improve the outcome [99]. In addition, a high risk of mild reduction in the verbal intellective quotient in 3-year-old children who were prenatally exposed to ClO_4_^−^ was observed irrespective of their mother’s thyroid function during pregnancy [100]. Furthermore, maternal ClO_4_^−^ concentration was found to positively correlate with male infant bodyweight, especially in preterm [101]. 

Several observations assessed the relationship between maternal perchlorate exposure and neonatal or infant thyroid homeostasis with controversial results according to the different clinical end-points used for the assessment of euthyroidism [102,103,104]. ClO_4_^−^ may affect children growth as reported by Mervish et al., who observed that girls with higher ClO_4_^−^ exposure displayed lower body mass index and waist circumference than controls [105]. In addition, the results of a cross-sectional study in 3151 participants (12–80 years old) displayed for each logarithmic unit increased exposure to both ClO_4_^−^ and thiocyanate, the level of free thyroxine decreased by 8% in adolescent girls and 9% in adolescent boys, respectively [106]. 

### 3.6. Overview on Other Halogenate Compounds 

Other halogenated compounds may interfere with I uptake as similarly observed for ClO_4_^−^, including bromine and brominated compounds [107] and fluoride and fluorinated compounds [108]. Bromine compounds naturally occur in marine and terrestrial plants, but industrial compounds account for 80% of bromine production [109]. In particular, bromine compounds are essentially found in phytochemical, pharmaceutics, pesticides, dyes, and photographic and water treatment chemicals [109]. Bromine has been found at higher concentrations in seawater (65 to 80 mg/L) compared to natural waters (in mean 0.5 mg/L) and groundwaters (1 to several mg/L) [110]. In addition, potassium bromate is an inexpensive oxidizing agent used as dough improver in the baking industry [111]. Specifically, it leads to the formation of disulfide bonds between gluten proteins, ameliorating bread’s proprieties, such as swelling and volume [112]. Chronic exposure to potassium bromate was associated with toxic effects and carcinogenicity in animal models [113,114,115]. However, no data are currently available to also confirm toxicity and carcinogenicity in humans, thus the International Agency for Research on Cancer classified potassium bromate in group 2B (possibly carcinogen to humans) [116]. Given these considerations, potassium bromate has been precautionarily banned from several countries, such as those in Europe, the United Kingdom, Canada, Nigeria, China, South Korea, and several countries in South America, but it is still considered safe in the United States. Indeed, according to the Food and Drug Administration, no sufficient evidence of potassium bromate adverse effects has been collected in humans thus allowing the use of additives in the bread baking industry not exceeding 75 parts per million [117]. For this reason, bromate levels should be constantly and reliably monitored in bread whether potassium bromate has been used as an additive in flour processing [112]. In one observational study in Nanchang (China), bromine was detected in all 131 whole blood samples, thus suggesting a higher prevalence of contamination among people [84]. The daily intake of bromide ranged from 2 to 8 mg in the USA and 9 mg in Europe (the Netherlands) [110]. Regulatory agencies defined limits of concentration bromide in drinking-waters at 6 mg/L for adults and 2 mg/L for children and acceptable daily intake currently ranges from 0 to 1 mg/Kg of bw [110]. Human exposure to brominated compounds usually occurs by food intake and consistently increases over time, resulting particularly higher in Occidental countries [118]. Breast milk as well as hair and adipose tissue may accumulate these chemicals, thus resulting as reservoirs for further persistence of brominated compounds in the human body [118]. Bromide may interfere with thyroid homeostasis, particularly competing with I uptake and I clearance [119,120] however, human toxicity data demonstrated that polybrominated compounds may interfere with gonadal function and sexual steroids’ metabolism [118]. Fluoride and fluorinated compounds has been found in different rock-forming minerals, fertilizers, pesticides, and propellants, and has also been found in drinking water generally at acceptable levels according to regulatory agencies (<1.5 mg/L or <4 mg/L) [121] and groundwater [122]. Considering that a low dose of fluoride increases overall oral health, several countries add it to their public water supply at 0.7 to 1.5 mg/L [118]. In Italy, public waters are naturally rich in fluoride (1 mg/L), thus making fluoride addition in public supply unnecessary [123]. However, fluoride concentration in public waters differs among regions, and is particularly higher in Lazio, where an excessive consumption of public drinking-water may lead to a fluoride overexposure [123]. Concerns over fluoride overexposure through drinking water have been raised in several countries [124], in which the levels of fluoride intake exceed safety limits, leading to a relevant increase in the prevalence of both dental and skeletal fluorosis [125]. Fluoride has been found to block I uptake by two fundamental mechanisms: inhibition of sodium-potassium ATPase and a cytokine-mediated reduction in NIS gene expression [126]. Indeed, fluoride exposure in early stages of life, mostly for preventing dental caries, is believed to be linked with an higher risk of future development of several diseases, including hypothyroidism and impaired intellective quotient [127]. Moreover, the exposure to fluoride concentration at 100 ppm (mg/L) in experimental conditions were associated with apoptosis, organelle damage and oxidative stress resulting in neurodegeneration, endocrine dysfunction and diabetes mellitus [128]. Due to anthropogenic and industrial activities, a great number of pollutant entry in water systems leading to possible concerns for wildlife and human health. Defluoridation of water may contribute in reducing the level of fluoride contamination in water and different physicochemical and electrochemical methods have been used for this purpose [129]. Among these, biosorption should be considered an easily available, recyclable and inexpensive tool [129].

## 4. Discussion

I sufficiency and euthyroidism are essential for preventing negative neurodevelopmental [130] outcomes and processing disorders [131], thus I deficiency or interference should be hazardous particularly during pregnancy and earlier stages of life given the particularly vulnerable thyroid function in this developmental phase. Conventionally, I deficiency has been defined as a 24-h urinary excretion <100 µg/L [132]. Given this criteria, more than 2 billion people worldwide are at high risk of iodine insufficiency and at least half of European citizens exhibit a mild to moderate I deficiency [133]. Italy has been historically defined as being endemic for I deficiency, particularly in the northern mountainous regions. Strategies of implementation of iodine intake have been allowed by law since 1972 through the use, on a voluntary basis, of fortified salt with an I content of 15 µg/Kg, subsequently augmented to 30 µg/kg (law 55/2005). Supplementation provided a slow but progressive improvement of iodine status over time but did not completely eradicate the risk [134,135] and the prevalence of mild-to-moderate I deficiency remains a current matter [136], especially in pregnancy and lactation [137]. Of note, patients with I deficiency should be considered as highly susceptible for developing I interference by food intake. ClO_4_^−^ has a short half-life (up to 8 h) due to a quick renal clearance [138], thus its accumulation in human body is clearly due to chronic exposure to drinks and food [139,140]. ClO_4_^−^ exposure may be harmful for thyroid homeostasis, especially in childhood and pregnancy. Two trials were performed to assess short term effects of a ClO_4_^−^ acute exposure (2 weeks) to either 0.5 or 3 mg daily, showing no effect on thyroid function [141,142]. However, 2 weeks of ClO_4_^−^ exposure at higher doses (10 to 30 mg per day) resulted in significantly reduced iodine uptake, potentially affecting thyroid hormone synthesis [143,144]. The results of these studies should be interpreted with caution, particularly considering that short-term exposure is usually insufficient to affect thyroid secretion of levothyroxine. Moreover, to achieve these levels of exposure, it could be necessary to have an extremely high daily consumption of ClO_4_^−^ for a limited period of time which is normally not reproducible in real life (i.e., 2 litres of drinking water at ClO_4_^−^ content as high as 200 µg/L). On the other hand, studies which assessed the effect of a chronic ClO_4_^−^ exposure (i.e., occupational) on thyroid hormone synthesis reported inclusive or equivocal results, despite I uptake being usually impaired in almost all participants [145,146,147]. Given these findings, it was difficult to make an unequivocal conclusion. The National Research Council of the National Academics sustained that, in healthy individuals, I uptake would be reduced by at least 75% for months in order to significantly impair thyroid hormone synthesis [47]. Thus, a sustained exposure to 0.5 mg/Kg of bw/day of ClO_4_^-^ would be most likely to induce a significant decline in I uptake consequentlyaffecting thyroid hormone synthesis [47]. However, the US Environmental Protection Agency adopted a recommended reference dose for ClO_4_^−^ at 0.7 µg/kg of bw/day [141]. This conservative decision was based upon a non-observed effect level found by Greer et al. in 2002 (7 µg/Kg of bw/day) divided for an uncertainty factor of 10 attributable to intra-human variability intended to calculate an acceptable daily intake [144]. The Office of Environmental Health Hazards Assessment developed a public health target for ClO_4_^−^ in drinking water of 6 µg/L in 2002 to 1 µg/L in 2012 [148]. In 2011, the Joint Food and Agriculture Organization—World Health Organization recognized a maximum tolerable daily intake of 10 µg/kg of body weight [149]. In Europe (France and Germany), the acceptable level of exposure to ClO_4_^−^ was set at 0.7 µg/kg of bw with a tolerable concentration in drinking water of 15 µg/L, successively reduced to 4 µg/L [47]. Furthermore, the European Food Safety Authority in 2014 predisposed the maximum tolerable daily dose of 0.3 µg/Kg of bw/day [70]. Water and soil contaminations have become a concern due to detrimental consequences for both wildlife and human health. Efficient methods for reducing the levels of ClO_4_^−^ in fruits and vegetables represent useful tools to decrease the levels of exposure. Considering that the contamination of fruits and vegetables should reflect ClO_4_^−^ concentration in soil, water for irrigation and fertilizers, several processes found application in this field [150]. As an example, ion exchangers, which replace ClO_4_^−^ with other resident anions, such as bicarbonate, sulfate and nitrate, are one of the most used methods for removing ClO_4_^−^ from water and may be considered as a tool for dropping ClO_4_^−^ levels throughout soil watering [151]. Biological degradation by perchlorate-reductase producer bacteria [152] or plants [150] could be counted as another useful method for reducing ClO_4_^−^ in water and soil, respectively. Photocatalytic reduction of aqueous oxyanions converts toxic anions (such as ClO_4_^−^ or bromate) into harmless and less and/or not toxic ions in contaminated waters [153]. However, several limitations have been described for this method, which include high costs of technologies, sunlight harvesting capability and generation of dangerous radical substances [153]. Physical methods include reverse osmosis coupled with nanofiltration membrane systems [154], or a less expensive semipermeable membrane system coupled with electrodialysis [155]. Moreover, iron-media adsorbent have been used for removing ClO_4_^−^ and other anions in aqueous solutions [156]. In particular, granular ferric hydroxide was found to induce a rapid uptake of ClO_4_^−^ in water, considering that its maximum absorption and equilibrium were achieved in 30 and 60 min, respectively at 25 °C with optimal pH at 3–7 [157]. ClO_4_^−^ contamination of soil and water is strictly related to geogenic ClO_4_^−^ naturally occurred in the atmosphere and subsequently precipitated. However, fertilizes may be considered a source of ClO_4_^−^ accumulated in food chain [158]. Among fertilizers, higher levels of ClO_4_^−^ were detected in nitrogenous fertilizers (32.6 mg/Kg) compared to natrium-phosphorus-potassium (12.6 mg/Kg), non-nitrogen (10.2 mg/Kg) and phosphates (11.5 mg/Kg) fertilizers [159]. Thus, the type and the amount of fertilizer may influence the source of entry for ClO_4_^−^ in crops. Additionally, agronomic practices of fertilization may also contribute in this risk. As an example, fertigation is an innovative and less expensive methods of fertilization which allows for less water being wasted, better distribution of fertilizers and superior micronutrient assimilation by crops, but given these principles, it may be easier to foster more significant accumulations of ClO_4_- in fruits and vegetables [60]. 

## 5. Conclusions

In conclusion, acute exposure to ClO_4_^−^ by food and drink should not be a harmful concern for thyroid homeostasis in healthy individuals. Generally, chronic exposure to ClO_4_^−^ by eating and drinking does not exceed the safety reference levels. However, lifelong effects of a low-dose exposure to ClO_4_^−^ are currently unknown and concerns remain, especially for highly susceptible individuals such as pregnant and breastfeeding women, infants and children, cigarette smokers and high vegetable consumers, such as vegans. These clusters of patients should be advised about this worry, and encouraged to limit daily consumption of rich in perchlorate vegetables, as well as to implement I intake.

On the other hand, producers should be encouraged to use specific culture systems, fertilizers (as an example nitrate-free) as well as technologies for reducing the level of ClO_4_^−^ in soil and irrigation waters in order to prevent an unnecessary ClO_4_^−^ enrichment of crops. For this purpose, economic sustainment should be considered particularly for small and medium-size companies in order to reduce management costs.

Finally, further and specific long-term studies are probably needed to better explore this issue, aiming to clarify whether monitoring of perchlorate exposure over time, especially in individuals at risk, could be of interest for endocrinologists for better defining strategies of detection and prevention in exposed patients.

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
