# Peer review of "Interference on Iodine Uptake and Human Thyroid Function by Perchlorate-Contaminated Water and Food"

_nutrients, 2020, doi:10.3390/nu12061669_

Round 1

Reviewer 1 Report

This review covers an important public health area least examined. The authors have done enormous work collecting this information. I have some few observations that need to be reconciled.

Main concerns

  1. Discussion line 301: The statement “exceed the limit of about 132%, 301 161% and 156% in adults, children and infants, respectively.” is incorrect. The values 132%, 301 161% and 156% are observed value reported in Vejdovszky et al article. The values are not reference limits. The authors must correct this misinformation.
  1. This review should also briefly mention the other halides or their derivatives that interfere with thyroid uptake of iodine. Such as fluoride and fluoridated compounds, and bromine and brominated compounds.
  2. Conclusions: If fruits and vegetables are sources of CLO4-, what suggestion do the authors have for fruits and vegetables producers to decrease levels?
  3. The written language needs to be significantly improved. There are too many grammatical errors, some of which I have indicated in my minor comments below.

Minor comments

  1. Abstract line 27, and introduction line 66: Authors need to spell out NIS on first use.
  2. Materials and Methods line 72. I think the word “secondary” should be “secondly”. However, since the authors have not used “firstly” the word “secondly” is out of place. It could be changed to “in addition” or “furthermore” or another suitable word.
  3. Section 3.3 line 136, “billion of people” should be “billion people”.
  4. Section 3.4 line 160, “were found into ice” should be corrected to “were found in ice”.
  5. Discussion line 274. “bw/die” should be corrected to “bw/day”
  6. Discussion line 301, The statement “exceed the limit of about 132%, 301 161% and 156% in adults, children and infants, respectively.” is incorrect.
  7. Discussion line 308. Please correct “assumption” to “consumption”

Author Response

May, 28th 2020

Dear Editor,

Authors thank reviewers for their comments and suggestions, and provide a point-by-point response as follow.

Reviewer 1

  1. Discussion line 301: The statement “exceed the limit of about 132%, 301 161% and 156% in adults, children and infants, respectively.” is incorrect. The values 132%, 301 161% and 156% are observed value reported in Vejdovszky et al The values are not reference limits. The authors must correct this misinformation.

We verified the reference. Vejdovszky et al analyzed the levels of perchlorate in vegetables in Austrian population, and estimated the level of exposure to perchlorate in case of hypothetical normal and higher consumption. The results of the study highlighted that a higher consumption of vegetable led to a relevant increase of the level of exposure regardless of age and this excess in levels of exposure were largely attributable to a great consumption of spinach. Moreover, they considered the tolerable daily intake (TDI) of perchlorate at 0.3 mcg/kg of body weight, as established by the European Food Safety Authority in 2014. Based upon this cut-off point, authors calculated that a hypothetical low exposure to perchlorate was safe, and did not exceed the TDI. Conversely, a high consumption of vegetables led to a relevant overcoming of TDI in all categories by 32% (adults), 61% (children) and 56% (infants).

  1. This review should also briefly mention the other halides or their derivatives that interfere with thyroid uptake of iodine. Such as fluoride and fluoridated compounds, and bromine and brominated compounds.

A novel paragraph has been created to briefly mention other halogenated compounds as you suggested. The paragraph entitled “3.6. Overwiev on other halogenate compounds” has been reported at page 6, and has been highlighted in yellow.

  1. Conclusions: If fruits and vegetables are sources of CLO4-, what suggestion do the authors have for fruits and vegetables producers to decrease levels?

In discussion, authors provided an overview on the main technologies employed for reducing levels of CLO4- in soil, irrigation waters and crops. In addition, difference in CLO4- content have been identified in several fertilizers and particularly those nitrogen-free normally had lower concentration of CLO4- and should be used for reducing the level of CLO4- contamination in crops.

  1. The written language needs to be significantly improved. There are too many grammatical errors, some of which I have indicated in my minor comments below.

Authors provide further modification of the text trying to improve language inaccuracies. However, if reviewer retains that this attempt to remove grammatical errors and text readability failed, we’ll request English Editing.

Minor comments

  1. Abstract line 27, and introduction line 66: Authors need to spell out NIS on first use.
  2. Materials and Methods line 72. I think the word “secondary” should be “secondly”. However, since the authors have not used “firstly” the word “secondly” is out of place. It could be changed to “in addition” or “furthermore” or another suitable word.
  3. Section 3.3 line 136, “billion of people” should be “billion people”.
  4. Section 3.4 line 160, “were found into ice” should be corrected to “were found in ice”.
  5. Discussion line 274. “bw/die” should be corrected to “bw/day”
  6. Discussion line 301, The statement “exceed the limit of about 132%, 301 161% and 156% in adults, children and infants, respectively.” is incorrect.
  7. Discussion line 308. Please correct “assumption” to “consumption”

All the modifications you suggested have been done.

PS: Since that the corresponding author email address has been wrongly reported in the previous version of the manuscript, please would you consider the correct email address as follows: [email protected].

Thank you for your consideration of this manuscript.

Sincerely,

Giuseppe Lisco, Anna De Tullio, Vito Angelo Giagulli, Giovanni De Pergola, Vincenzo Triggiani

Reviewer 2 Report

In the review “Interference on iodine uptake and human thyroid function by perchlorate-contaminated water and food” Lisco G and colleagues give an overview of the effect of perchlorate exposure on thyroid disorders. The manuscript is interesting and clearly presented, taking also into account  strengths and limitations of literature data.

Although the text is well written, in my opinion  some paragraphs are too detailed  and often placed in a wrong way. I think that the review could be more attractive if authors better re-organized single paragraphs. I would suggest to shorten the part concerning iodine metabolism, uptake and deficiency, since so many details are  unnecessary considering  the aim of the review. Moreover, it would be more appropriate to move parts from the  discussion to the main text. For example, paragraph 3.4 could include perchlorate levels in food and water described into discussion (from line 284); the same is true for  the first part of  discussion (perchlorate toxicity ), that could be inserted more appropriately in a new paragraph starting from line 179 . My last concern relies on the final part, which seems barely discussed. The authors are required to improve it in order to be in line with the conclusions. 

Author Response

May, 28th 2020

Dear Editor,

Authors thank reviewers for their comments and suggestions, and provide a point-by-point response as follow.

Reviewer 2

Although the text is well written, in my opinion some paragraphs are too detailed and often placed in a wrong way. I think that the review could be more attractive if authors better re-organized single paragraphs. I would suggest to shorten the part concerning iodine metabolism, uptake and deficiency, since so many details are unnecessary considering the aim of the review. Moreover, it would be more appropriate to move parts from the discussion to the main text. For example, paragraph 3.4 could include perchlorate levels in food and water described into discussion (from line 284); the same is true for the first part of discussion (perchlorate toxicity), that could be inserted more appropriately in a new paragraph starting from line 179. My last concern relies on the final part, which seems barely discussed. The authors are required to improve it in order to be in line with the conclusions. 

As reviewer suggested, authors re-organized every single paragraph highlighting every single title of the modified paragraphs. Paragraphs dedicated to the description of iodine metabolism, uptake and deficiency have been replaced by a single resuming paragraph (see paragraph 3.1). A more detailed description of perchlorate interference, food and water and perchlorate toxicity have been also reported. Discussion and conclusion have been enriched also in light of other received suggestions.

PS: Since that the corresponding author email address has been wrongly reported in the previous version of the manuscript, please would you consider the correct email address as follows: [email protected].

Thank you for your consideration of this manuscript.

Sincerely,

Giuseppe Lisco, Anna De Tullio, Vito Angelo Giagulli, Giovanni De Pergola, Vincenzo Triggiani

Round 2

Reviewer 1 Report

The manuscript has improved significantly. I have some few minor suggested edits.

  1. Abstract, line 22: “Natrium/Iodide Symporter (NIS)”. Please change to “Natrium-Iodide Symporter (NIS)” because the "/" also means "or", however, sodium and iodine are symported together from the apical side to the cytosolic side of the thyroid cells.
  2. Materials and methods, line 78: Please change to natrium-iodine symporter per my earlier comment.
  3. Section 3.6. Line 257: Edit “Overwiev” to “Overview” to correct it.
  4. Section 3.6. Overview on other halogenate compounds: For line 259, edit to correct “fluiride” to “fluoride”.
  5. Section 3.6. Overview on other halogenate compounds: For lines 259-262, also note that the use of potassium bromate in bread baking, and excessive ingestion of fluoride from some drinking water make them potential sources of bromide and fluoride, respectively.

Section 3.6. Overview on other halogenate compounds: For line 284: Chane “(3)natrium/(2) potassium ATPase” to “natrium-potassium ATPase” to conform to its use in most scientific literature notation of Na-K ATPase. Since readers are more familiar with sodium-potassium ATPasse, you can use “sodium-potassium ATPasse” to describe this enzyme

Author Response

May, 31th 2020

Dear Editor,

Authors thank reviewers for their comments and suggestions, and provide a point-by-point response as follow.

Reviewer

  1. Abstract, line 22: “Natrium/Iodide Symporter (NIS)”. Please change to “Natrium-Iodide Symporter (NIS)” because the "/" also means "or", however, sodium and iodine are symported together from the apical side to the cytosolic side of the thyroid cells.
  2. Materials and methods, line 78: Please change to natrium-iodine symporter per my earlier comment.
  3. Section 3.6. Line 257: Edit “Overwiev” to “Overview” to correct it.
  4. Section 3.6. Overview on other halogenate compounds: For line 259, edit to correct “fluiride” to “fluoride”.

All the modifications you suggested have been done. Of note, any modifications have been marked in red.

Section 3.6. Overview on other halogenate compounds: For lines 259-262, also note that the use of potassium bromate in bread baking, and excessive ingestion of fluoride from some drinking water make them potential sources of bromide and fluoride, respectively.

As you suggested, authors reported some aspects linked to the use of potassium bromate in flours for improving bread characteristics. Of note, in Europe as well as in other countries worldwide, the use of potassium bromate has been banned because of concerns inherited form animal models (thyroid and renal toxicity and carcinogenesis). On the other hand, FDA considered as insufficient the evidences of bromate toxicity in humans, thus allowing the use of potassium bromate as an additive in bread baking industry. Given this choice, authors believed that an adequate surveillance on brominate compounds in food should be useful in order to avoid an unnecessary overexposure. Skeletal fluorosis is a serious disease commonly due to an excessive fluoride exposure by drinking water. This condition is critical particularly for countries in which water fluoride concentrations exceeded the level of safety (1 mg/l). Considering that skeletal alterations are irreversible and difficult to treat once they occurred, to avoid public health concerns is mandatory to prevent an excessive exposure to fluoride by water defluorination.

Section 3.6. Overview on other halogenate compounds: For line 284: Chane “(3)natrium/(2) potassium ATPase” to “natrium-potassium ATPase” to conform to its use in most scientific literature notation of Na-K ATPase. Since readers are more familiar with sodium-potassium ATPasse, you can use “sodium-potassium ATPasse” to describe this enzyme

All the modifications you suggested have been done. Of note, any modifications have been marked in red.

PS: Since that the corresponding author email address has been wrongly reported in the previous version of the manuscript, please would you consider the correct email address as follows: [email protected].

Thank you for your consideration of this manuscript.

Sincerely,

Giuseppe Lisco, Anna De Tullio, Vito Angelo Giagulli, Giovanni De Pergola, Vincenzo Triggiani